# SHED-CM: The Safety and Efficacy of Conditioned Media from Human Exfoliated Deciduous Teeth Stem Cells in Amyotrophic Lateral Sclerosis Treatment: A Retrospective Cohort Analysis

**DOI:** 10.3390/biomedicines12102193

**Published:** 2024-09-26

**Authors:** Yasuhiro Seta, Konomi Kimura, Goto Masahiro, Kimiko Tatsumori, Yasufumi Murakami

**Affiliations:** 1Hitonowa Medical, Chiyoda 102-0085, Japan; 2Kokoro to Karada Clinic, Ichikawa 272-0133, Japan; 3Sanyukai Medical Corporation, Chuo 104-0061, Japan; 4Department of Biological Science and Technology, Faculty of Advanced Engineering, Tokyo University of Science, Noda 278-8510, Japan

**Keywords:** ALS, SHED-CM, regenerative medicine, neurodegeneration, stem cells

## Abstract

Background/Objectives: Amyotrophic lateral sclerosis (ALS) is a progressive and irreversible neurodegenerative disease with limited treatment options. Advances in regenerative medicine have opened up new treatment options. The primary and exploratory objectives of this retrospective cohort study were to evaluate the safety and efficacy of stem cells from human exfoliated deciduous teeth-conditioned media (SHED-CM). Methods: Safety assessments included adverse events, vital signs, and laboratory test changes before and after administration, and efficacy was measured using the ALS Functional Rating Scale-Revised (ALSFRS-R), grip strength, and forced vital capacity in 24 patients with ALS treated at a single facility between 1 January 2022, and 30 November 2023. Results: While ALSFRS-R scores typically decline over time, the progression rate in this cohort was slower, suggesting a potential delay in disease progression. Alternatively, improvements in muscle strength and mobility were observed in some patients. Although adverse events were reported in only 3% of cases (no serious allergic reactions), the treatment-induced changes in vital signs and laboratory results were not clinically significant. Conclusions: The SHED-CM treatment is a safe and potentially effective therapeutic option for patients with ALS. Further research is needed to optimize the SHED-CM treatment; however, this study lays the groundwork for future exploration of regenerative therapies for ALS.

## 1. Introduction

Amyotrophic lateral sclerosis (ALS) is an incurable disease characterized by selective and progressive degeneration and loss of both upper and lower motor neurons. The disease progresses rapidly, with an average survival time from onset to death or the need for respiratory support of 24–50 months [1]. Although the cause of ALS is still not completely understood, it has been reported that approximately 10% of patients have a family history of ALS, and approximately 20% of ALS patients have mutations in the SOD1 gene [2]. Abnormalities in free radical processing, glutamate toxicity, and mitochondrial dysfunction have been identified as possible causes of sporadic ALS [3]. Furthermore, abnormalities in TDP-43 cause the protein to form abnormal aggregates, inducing nerve cell death and leading to ALS onset [4].

Currently, three drugs—riluzole, edaravone, and tofersen—have received accelerated FDA approval for ALS treatment [5]. Although these drugs counteract glutamate toxicity and free radicals, their therapeutic efficacy is limited, and they can only slow disease progression to tracheostomy for 3–6 months [6,7]. New treatments continue to be developed, including perampanel [8], hepatocyte growth factor (HGF) [9], methylcobalamin [10], bosnitib [11], and ropinirole [12], which do not improve symptoms but rather slow disease progression. As the cause of ALS is not completely understood and multiple factors may be involved, treatment with a single drug may be difficult.

Stem cell therapy is a promising treatment for the complex pathology of ALS [13]. This therapy can influence various pathogenic mechanisms through nutritional support and immunomodulation, potentially slowing disease progression. Several case reports have suggested the potential efficacy of adipose-derived mesenchymal stem cells (ADSCs) [14,15,16,17]. However, the effectiveness of ALS in the treatment of ADSC remains unclear. Although a Cochrane review assessed the feasibility and safety of cell-based therapies compared with placebo or no treatment in patients with ALS/motor neuron disease (MND), current evidence suggests that ADSCs and bone marrow-derived mesenchymal stem cells do not support the use of lineage stem cells (MSCs) [18]. Despite showing certain advantages, adipose and bone marrow-derived MSC transplantations suffer from limited data collection and high costs. However, mesenchymal stem cells derived from deciduous teeth (SHED) have distinct properties and are considered potential therapeutics for ALS.

SHEDs are localized in the pulp of baby teeth [19]. They are pluripotent and can differentiate into various cell types, including neurons. SHEDs are an ideal cell source for stem cell therapy because of their neuroprotective and immunomodulatory properties [20]. Obtaining primary teeth is also advantageous from an ethical and safety perspective and is less problematic than other adult stem cell sources [21]. However, SHEDs are associated with immune response-associated issues, as these cells are donated by other individuals. To address this, using a serum-free conditioned culture medium (SHED-CM), which collects the components secreted from SHEDs, is considered effective. By utilizing these secreted components rather than the cells themselves, SHED-CM is expected to bypass any potential immune reactions, enable cell recycling during the manufacturing process, and stabilize the quality and supply.

SHEDs have been shown to release numerous factors, including neurotrophic factors, that promote recovery from central nervous system disorders [22]. SHED-CM [23] contains components such as vascular endothelial growth factor (VEGF) and HGF, which are effective against neurological diseases, including ALS, at much higher levels than bone marrow-derived stem cell CM [24]. A recent study showed that SHED-CM significantly suppresses mutant SOD1-induced intracellular aggregation and neurotoxicity and has a protective effect on motor neurons derived from induced pluripotent stem cells generated from patients with ALS [25]. Additionally, SHED-CM has been used in ALS mouse models and has been reported to extend the survival period of mice [26]. Therefore, SHED-CM represents a promising treatment option for patients with ALS. In this study, we retrospectively analyzed the experience of administering SHED-CM to patients with ALS at our hospital.

## 2. Materials and Methods

This was a retrospective cohort study that included patients diagnosed with ALS who visited our clinic between 1 January 2022 and 30 November 2023. We enrolled patients with ALS who received SHED-CM within this period, were aged 16 to 90 years, had a score of 12 or higher on the ALS Functional Rating Scale-Revised (ALSFRS-R), and had not undergone tracheostomy. Patients who did not provide their consent to participate in this study were excluded. All data were evaluated for safety. Several patients experienced difficulties presenting to the hospital due to the progression of their ALS; therefore, we evaluated the efficacy of the drug from the first to the 12th dose, which was administered consecutively, and for which all patients presented to the hospital.

### 2.1. Data Collection

We collected the following data from past medical records:

Basic information: age, sex, medical history, complications, presence of allergies;

Body measurements: height, weight;

Vital signs: body temperature, systolic and diastolic blood pressure, pulse, and SpO_2_. These results were measured twice: at the time of the hospital visit and after the end of the infusion;

Signs at the first and fourth visits after infusion administration;

Clinical test data: hematological tests, blood biochemistry tests, coagulation system tests, ALSFRS-R score, grip strength, and respiratory function;

Scope of data usage.

### 2.2. Isolation of Stem Cells from Human Baby Teeth

SHEDs were isolated from children aged 6–12 years following the appropriate donor criteria and ethical standards. The key properties were analyzed using flow cytometry, which confirmed the expression of a series of MSC markers (CD90, CD73, and CD105) and no endothelial or hematopoietic markers (CD34, CD45, CD11b/c, or HLA-DR). These cells exhibited adipogenic, chondrogenic, and osteogenic differentiation abilities.

### 2.3. Preparation of SHED-CM

SHEDs were washed with phosphate-buffered saline (PBS) when they reached 70–80% confluence at passage numbers 3 to 9. Subsequently, the culture medium was changed to a serum-free medium (DMEM). After 48 h of incubation, the medium was collected, sterilized, and frozen. These processes were performed by U-factor Co., Ltd., Chiyoda-ku, Japan and the resulting SHED-CM was named “U-factor^®^“. The product was delivered to the clinic in a frozen state.

### 2.4. Statistical Analysis

The collected data were analyzed using the Python software (3.10.12). Evaluation of treatment efficacy included pre- and post-dose safety evaluations, changes in blood test results, changes in ALSFRS-R scores, and analysis of other laboratory test results. A two-tailed test with a *p*-value < 0.05 was used to determine significance. Blanks and clinically anomalous outliers were excluded from the analysis.

### 2.5. Safety Evaluation

Safety was evaluated using the administration data. This dataset included changes in vital signs (temperature, pulse, blood pressure, and SpO_2_) before and after treatment, changes in blood test results, and the presence or absence of treatment-related adverse events. At each visit, changes in vital signs before and after IV administration were compared using paired *t*-tests. Additionally, changes in vital signs and blood test results between the first and fourth IV administrations were analyzed using a paired *t*-test.

### 2.6. Effectiveness Evaluation

Because some patients experienced difficulties presenting to the hospital due to ALS progression, the efficacy was evaluated using data from the first 12 administration sessions, limited to patients who were able to visit the hospital consecutively. Discrepancies in the initial ALSFRS-R scores, grip strength, and forced vital capacity made it difficult to uniformly analyze all patients. Therefore, we used it for the first time as a base and evaluated the subsequent changes and improvement rates.

### 2.7. Information of SHED-CM

Liquid chromatography–mass spectrometry (LC–MS) was performed by U-factor Co., Ltd. to identify the components contained in SHED-CM. LC–MS is an analytical technique that combines the physical separation capabilities of liquid chromatography with the mass analysis capabilities of mass spectrometry. It is commonly used to identify and quantify components of complex mixtures, such as proteins, peptides, and metabolites, by separating them based on their chemical properties and analyzing their mass-to-charge ratios.

### 2.8. Ethics

This study was approved by the Ethics Committee of the Japan Registry of Clinical Trials (jRCT1031230719). An opt-out method was adopted for research participation, and written informed consent was obtained from the patients who visited the hospital. Particular attention was given to the protection of patient privacy and personal information, and patient dignity was respected throughout the study.

## 3. Results

### 3.1. Result Summary

A total of 24 (mean age of 55.2 years) patients were treated with SHED-CM between 1 January 2022 and 30 November 2023. Of these, 15 (62.5%) were men, and nine (37.5%) were women. When dividing by ALS type, 13 patients (54.2%) presented the upper limb type, six (25%) presented the lower limb ALS, and five (20.8%) presented the bulbar type. Twenty patients (83.3%) were treated with riluzole or radicut (Table 1).

### 3.2. Treatment Content

Patients received an average of 123 mL (80–150 mL/patient) of SHED-CM per visit and were observed approximately every week. Based on the first visit, the fourth, eighth, and twelfth doses were, on average, administered 21.5, 60.7, and 95.5 days later, respectively.

### 3.3. Safety Results

Our assessment of the changes in vital signs before and after infusion at each visit revealed significant changes in blood pressure and pulse rate (Table 2). Vital signs were recorded upon arrival at the hospital, and intravenous infusion was administered for 1 h. Although most changes might be explained by the bed rest period, the influence of SHED-CM administration cannot be completely ruled out. Additionally, no significant differences in vital signs were observed between the first and fourth infusion administrations (Table 3). However, blood sample analyses before the first and fourth infusions revealed a significant difference in the mean corpuscular hemoglobin content (MCH) and mean corpuscular hemoglobin concentration (MCHC) (Table 4). Notably, these mean differences were negligible, and the possibility of measurement errors must be considered. Even if these changes are significant, they are unlikely to be clinically significant. Lastly, although changes in total protein (TP) were observed, this is thought to be mainly due to measurement errors such as decreased food intake due to ALS pathology.

### 3.4. Adverse Event

Adverse events were observed in 10 patients (approximately 3%) during a total of 314 infusion administrations (Table 5). No serious allergic reactions, including delayed reactions, were observed. Adverse events were minor and included fever, headache, diarrhea, skin rash, itching, and high D-dimer levels.

### 3.5. Effectiveness Assessment

The initial ALSFRS-R score and the changes in score after the fourth, eighth, and twelfth administrations are presented in Figure 1. The participants were categorized by sex into three groups: those whose scores improved, those who remained the same, and those who deteriorated. Women showed a greater improvement than men (*p* = 0.08) (Table 6). The average rate of change in the score after each administration was calculated and evaluated (Figure 2). The average score change at the fourth administration was 0.0 points (standard error (SE) = 0.32), whereas those at the eighth and twelfth administrations were −0.21 (SE = 0.81) and −1.77 (SE = 1.35), respectively. A detailed analysis of individual patient data revealed cases where the score remained unchanged, as well as where the score improved. For forced vital capacity and grip strength, several data points were missing, making overall evaluation challenging.

### 3.6. Information of SHED-CM

LC–MS identified more than 1700 proteins in SHED-CM. A subset of these proteins is shown (Table 7).

## 4. Discussion

In this study, safety and efficacy were evaluated by administering SHED-CM to ALS patients. Although no serious adverse events were observed following administration, the changes in vital signs and blood results were mild. Vital signs remained stable over the past month, indicating no immediate concerns. These results were derived from a *t*-test, which cannot entirely rule out the possibility of a significant difference. Therefore, future evaluations should consider equivalence testing to provide a more comprehensive assessment.

The incidence of adverse events was 3%, with high D-dimer levels potentially attributed to ALS pathology. Hence, we estimate that the number of adverse events that are directly related to the treatment is even lower. Considering that the incidence of adverse events in cell therapy is approximately 3–6% [27,28], the incidence of adverse events in this study is comparable to, or even lower than, that of cell therapy. Furthermore, in cell therapy, MSCs have been associated with surface markers and stromal factors that promote blood clot formation [29], suggesting that cell-free SHED-CM may reduce the risk of thrombus formation.

The evolution of ALSFRS-R scores is particularly noteworthy. The progression of ALS differs from patient to patient, making general assessments challenging. Nevertheless, if we refer to the change in ALSFRS-R scores for the placebo group in Phase III trials of investigational drugs [10], ALS progresses by approximately one point per month. This suggests that SHED-CM administration in this study may delay the progression of ALS, although it is a comparison with a randomized placebo group. Additionally, although ALS is usually a progressive and irreversible disease, some patients in this study showed improvements in symptoms such as leg movement, ease of walking, and arm movement. These improvements contributed to an increase in the ALSFRS-R scores. Symptoms improved in the upper motor neuron damage (improvement in spasticity) and lower motor neuron damage (improvement in weakness) groups. These results suggest that SHED-CM administered from peripheral blood may promote neuroprotection and regeneration by crossing the blood-nerve barrier (BNB), including the blood–brain barrier (BBB). It is possible that the cytokines in the SHED-CM, which can cross the BNB, act directly. Alternatively, the immune regulatory effects within the body might act indirectly by passing through the BNB. In these cases, we published videos on our clinic’s website showing examples of specific improvements. Regarding the grip strength results and respiratory function tests, we plan to perform a more detailed analysis and clarify the results by including more cases in the future.

Regenerative medicine is a field of medicine focused on treatments using cells, including stem cell culture supernatant treatments [30,31]. Previous studies have reported no major side effects when SHED-CM was administered to mice [32]. However, because SHED-CM is a liquid with complex components, defining and calculating its dosage is challenging. For conventional pharmaceuticals, dosage is often calculated from the body surface area of the mouse (body surface area method). However, this method’s direct application to SHED-CM is challenging.

In the field of cell therapy, several treatments are being conducted worldwide, and protocols such as the number of cells to be administered are also being explored. For example, some clinical trials administer approximately 100 to 200 million cells [28,33], and these studies provide accumulating evidence of safety. Recent studies have shown that although stem cells may have therapeutic effects, most of the administered cells are trapped in the lungs and spleen and are processed before reaching their target organs [34,35]. However, this approach may still be clinically effective. This means that either a small number of stem cells reach the target organ, engraft, and exert the observed improvements or that the cytokines and exosomes secreted by the trapped cells directly improve the target organ or regulate the immune system (indirect effect) [31]. Specifically, it may indirectly improve efficacy by shifting macrophages/microglia from M1 to M2 anti-inflammatory cells [36]. Therefore, SHED-CM is valuable as a collection of components secreted from cells, creating a simulated environment outside the body rather than directly performing cell transplantation. Previous animal experiments using SHED-CM have suggested the possibility of neural regeneration not only in ALS models but also in cerebral infarction and dementia models [37,38].

Given the hundreds of different cytokines and exosomes, identifying their individual effects is challenging. However, equivalence with cell therapy can be achieved using an approach similar to that of cell administration.

Specifically, we used SHED-CM obtained from approximately 200,000 cells/mL, with a dose per administration of 120 mL, indicating a total of approximately 24 million cells. This dose is equivalent to approximately 12–25% of the 100–200 million cells normally used in cell transplantation and is thought to be safe.

More than 1700 protein components were detected in SHED-CM using liquid chromatography–mass spectrometry. SHED-CM contains components such as VEGF and HGF, which are known to be effective against neurological diseases, including ALS, and cytokines, such as MCP-1, which promote M1/M2 shift changes (Table 6). We propose that these and other components of SHED-CM promote neuroprotection and regeneration and that an improvement in symptoms was observed. We believe that it is important to deepen our understanding of SHED-CM treatment and further clarify its role in the treatment of ALS by evaluating the active ingredients and their combinations.

### Limitations

This was a retrospective study, and owing to its nature, further consideration of the quality and evaluation of the data is required. In particular, it was not possible to obtain high-quality data for some measurements, such as spirometry and grip strength. The quality of the data was also affected by the fact that the study was conducted at a single institution, and the frequency of patient visits was irregular. These factors should be considered when generalizing the results. In future research, prospective studies involving other facilities should be conducted to obtain more accurate data and strengthen the results of this study. This will enable a more reliable evaluation of the safety and efficacy of the SHED-CM treatment.

Finally, although the efficacy results of this study are preliminary, they suggest potential benefits that require further investigation. We emphasize that the efficacy findings are exploratory in nature and suggest future studies with larger cohorts to validate these results.

## 5. Conclusions

The results of this study suggest that the administration of SHED-CM may provide new hope for the treatment of ALS. ALS is generally considered a progressive and irreversible disease. However, in this study, improvements in symptoms were observed in some cases, which represents a promising preliminary finding. In the future, we hope that, by accumulating further data and identifying the cytokines present in SHED-CM, we will be able to provide more effective treatments for patients with ALS.

## Figures and Tables

**Figure 1 biomedicines-12-02193-f001:**
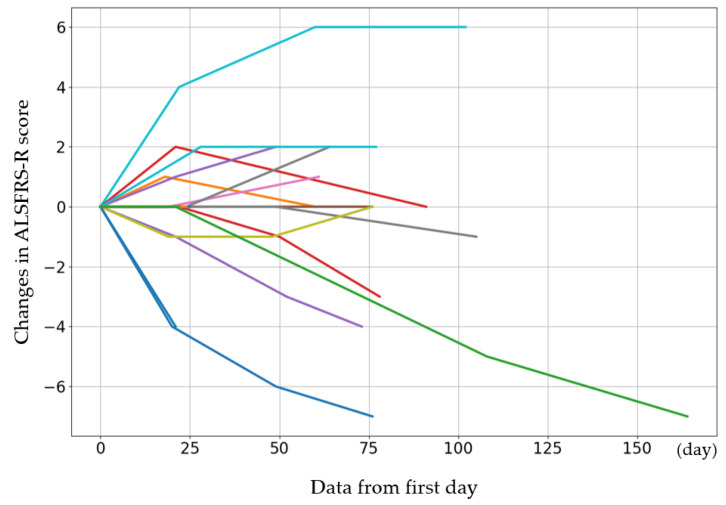
The initial ALSFRS-R score for each case and the changes in score after the 4th, 8th, and 12th administrations. The cases show increases or maintenance in scores. The colors are used to distinguish individual cases and do not represent differences in variables.

**Figure 2 biomedicines-12-02193-f002:**
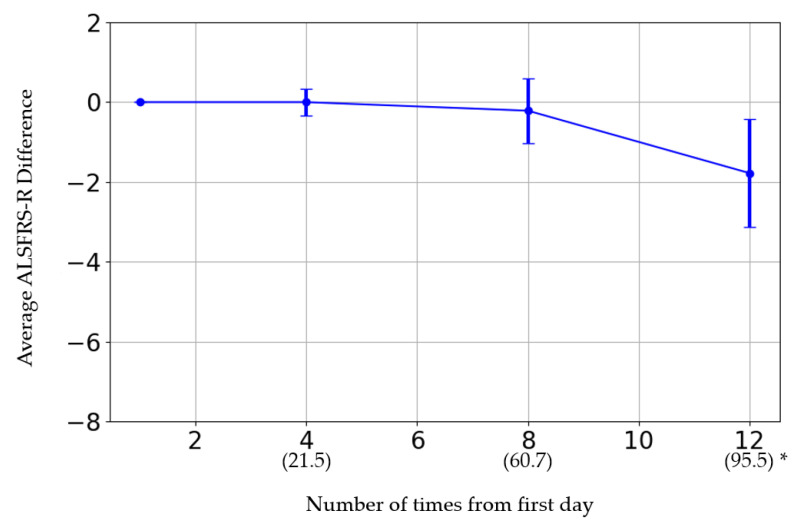
* The horizontal axis represents the number of administrations. The date in parentheses is from the first administration date.

**Table 1 biomedicines-12-02193-t001:** Baseline demographic and clinical characteristics (full analysis set).

Characteristic	Total (N = 24)
Age, median (range), yr.	55.2 (38–81)
Male sex, n (%)	15 (62.5)
Disease type, No. (%)	
Upper extremity	13 (54.2)
Lower extremity	6 (25)
Bulbar	5 (20.8)
ALS type, No. (%)	
Familial ALS	1 (4.1)
Sporadic ALS	23 (95.8)
Concomitant use of riluzole or edaravone, No. (%)	20 (83.3)
ALS severity at baseline, No. (%) *	
Grade 1	6 (25)
Grade 2	8 (33)
Grade 3	7 (29)
Grade 4	3 (12.5)
ALSFRS-R score at first (range)	34.6 (18–47)
Amount of SHED-CM (range)	123.7 (80–150)

Abbreviations: ALS, amyotrophic lateral sclerosis; ALSFRS-R, Revised Amyotrophic Lateral Sclerosis Functional Rating Scale. * ALS severity: The severity of ALS symptoms was graded according to the Japan ALS severity classification of grades 1 to 5, with grade 5 being the most severe.

**Table 2 biomedicines-12-02193-t002:** Safety outcomes (comparison before and after administration on the day).

Vital Sign	Mean (SE)		Difference (95% CI)	*p*-Value
	Before	After		
Body temperature (°C)	36.4 (0.02)	36.4 (0.02)	0.01 (−0.04 to 0.05)	0.74
Systolic blood pressure (mmHg)	128.2 (0.87)	126.9 (0.91)	1.33 (0.14 to 2.66)	0.04
Diastolic blood pressure (mmHg)	81.2 (0.65)	78.8 (0.68)	2.4 (1.33 to 3.46)	<0.001
Pulse (bpm)	81.2 (0.74)	74.8 (0.69)	6.46 (5.5 to 7.42)	<0.001
SpO_2_ (%)	96.1 (0.1)	96.2 (0.13)	−0.14 (−0.37 to 0.09)	0.23

Our assessment of the changes in vital signs before and after treatment at each visit. There was a statistically significant difference in diastolic blood pressure (mmHg) and pulse (bpm) (*p* < 0.001).

**Table 3 biomedicines-12-02193-t003:** Safety outcomes (comparison before the first and fourth dose).

Vital Sign	Mean (SE)		Difference (95% CI)	*p*-Value
	1st	4th		
Body temperature (°C)	36.4 (0.07)	36.3 (0.08)	0.13 (0.04 to 0.5)	0.03
Systolic blood pressure (mmHg)	128.5 (2.54)	126.9 (3.1)	1.63 (−6.17 to 13.72)	0.41
Diastolic blood pressure (mmHg)	82.7 (2.32)	82.3 (2.22)	0.38 (−13.05 to 8.16)	0.61
Pulse (bpm)	82.7 (2.51)	84.4 (2.36)	−1.71 (−12.36 to 11.25)	0.92
SpO_2_ (%)	96.6 (0.39)	96.1 (0.53)	0.5 (−1.44 to 1.89)	0.77

Our assessment of the changes in vital signs between the first and fourth infusion administrations. There was no significant difference.

**Table 4 biomedicines-12-02193-t004:** Safety outcomes (comparison before the first and fourth dose).

Blood Collection Items	Mean (SE)		Difference (95% CI)	*p*-Value
	1st	4th		
Mean corpuscular hemoglobin (MCH) (pg)	30.6 (0.44)	30.3 (0.43)	0.31 (0.17 to 0.44)	0.004
Mean corpuscular hemoglobin concentration (MCHC) (%)	33.1 (0.2)	32.8 (0.23)	0.32 (0.09 to 0.55)	0.04
Total protein (TP) (g/dL)	7.4 (0.1)	7.1 (0.11)	0.24 (0.1 to 0.37)	0.02

Blood sample analyses before the first and fourth infusions. There was a statistically significant difference in MCH, MCHC, and TP (*p* < 0.05).

**Table 5 biomedicines-12-02193-t005:** Adverse Events.

Variable	Number of Cases (%)
Fever grade 0	3 (3%)
Headache grade 1	3 (3%)
Diarrhea grade 1	1 (1%)
Maculopapular eruption grade 1	1 (1%)
Itching sensation grade 2	1 (1%)
D-dimer high value grade 2	1 (1%)
Total (314 administrations)	10 (3.1%)

**Table 6 biomedicines-12-02193-t006:** ALS Progression by Sex.

Sex	Improvement	Maintenance	Deterioration
Female, n	2	5	2
Male, n	5	2	8

**Table 7 biomedicines-12-02193-t007:** LC–MS data for SHED-CMs (provided by U-factor).

Protein. IDs	Gene Names	Protein Names	Intensity (log2)
P15692	*VEGFA*	Vascular endothelial growth factor A	16.38
P49765	*VEGFB*	Vascular endothelial growth factor B	13.83
P49767	*VEGFC*	Vascular endothelial growth factor C	15.49
P14210	*HGF*	Hepatocyte growth factor	13.48
P13500	*CCL2/* *MCP-1*	C-C motif chemokine 2/ Monocyte Chemoattractant Protein-1	15.38
P23560	*BDNF*	Brain-derived neurotrophic factor	12.59
P01138	*NGF*	Beta-nerve growth factor	14.30
P05019	*IGF1*	Insulin-like growth factor I	13.62

## Data Availability

Data available in a publicly accessible repository. The original data presented in the study are openly available in FigShare at https://figshare.com/articles/dataset/dx_doi_org_10_6084_m9_figshare_6025748/6025748 (acessed on 8 September 2024).

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
