# Peer review of "SHED-CM: The Safety and Efficacy of Conditioned Media from Human Exfoliated Deciduous Teeth Stem Cells in Amyotrophic Lateral Sclerosis Treatment: A Retrospective Cohort Analysis"

_biomedicines, 2024, doi:10.3390/biomedicines12102193_

Round 1
Reviewer 1 Report (Previous Reviewer 1)
Comments and Suggestions for Authors
The authors have responded to my suggestions in a satisfactory manner. Specifically, they indicated in the text that the study findings are preliminary and require further investigation.
Author Response
Dear Reviewer,
Thank you for your positive feedback. We appreciate your valuable suggestions, and we are pleased that our revisions have addressed your concerns.
Sincerely,
Yasuhiro Seta
Reviewer 2 Report (Previous Reviewer 2)
Comments and Suggestions for Authors
I appreciate the authors for the improvements they have made.
Author Response
Dear Reviewer,
Thank you for your positive feedback. We appreciate your valuable suggestions, and we are pleased that our revisions have addressed your concerns.
Sincerely,
Reviewer 3 Report (Previous Reviewer 3)
Comments and Suggestions for Authors
Authors response the reviewer’s comments. But there are still several problems in this study.
Authors mention that “Due to the limited resources and the exploratory nature of this initial study, same Dr was responsible for both clinical evaluation and data analysis”. It is difficult to accept this comment, because there is no independence between clinical evaluation and data analysis.
In addition, there is no CRC in this study. Does only one doctor explain the study design, informed consent, clinical evaluation and data analysis? If so, this study is not acceptable as authorized clinical study.
Author Response
comment"Authors mention that “Due to the limited resources and the exploratory nature of this initial study, same Dr was responsible for both clinical evaluation and data analysis”. It is difficult to accept this comment, because there is no independence between clinical evaluation and data analysis.
In addition, there is no CRC in this study. Does only one doctor explain the study design, informed consent, clinical evaluation and data analysis? If so, this study is not acceptable as authorized clinical study."
Response
Dear Reviewer,
Thank you for your valuable feedback. We would like to provide additional clarification regarding your concern about the independence between clinical evaluation and data analysis in our retrospective study.
-
Fixed Nature of Data in a Retrospective Study:
Since this is a retrospective study, the data used for analysis were already collected and are fixed. As a result, we believe that a Clinical Research Coordinator (CRC) is not necessarily required in this case. Any missing or incomplete data were excluded from the analysis, as mentioned in the 2.4 Statistical Analysis section: "Blanks and clinically anomalous outliers were excluded from the analysis." -
Independence Between Evaluator and Data Analyst:
While having separate individuals for clinical evaluation and data analysis is ideal in ensuring independence, we believe that in the context of a retrospective study, it is acceptable for the same person to perform both roles. This is supported by the fact that the STROBE Statement (https://www.strobe-statement.org/strobe-publications/), a widely recognized guideline aimed at improving the reporting of observational studies in epidemiology, does not specifically require separate evaluators and analysts.However, to further ensure objectivity, we had another physician independently review and verify the data, and we have acknowledged their contribution in the manuscript's Acknowledgments section.
We hope this clarification addresses your concerns. We remain open to any further suggestions you may have and appreciate your consideration.
Sincerely,
Yasuhiro Seta

Round 2
Reviewer 3 Report (Previous Reviewer 3)
Comments and Suggestions for Authors
Authors emphasize that this is "retrospective study". But they treated 24 ALS patients with SHED-CM at a single facility between January 1, 2022, and November 30, 2023. I wonder if the clinical trial system in the clinic is appropriate or not.
This manuscript is a resubmission of an earlier submission. The following is a list of the peer review reports and author responses from that submission.
Round 1
Reviewer 1 Report
Comments and Suggestions for Authors
The researchers used media conditioned stem cells from human exfoliated deciduous teeth to treat symptoms of amyotrophic lateral sclerosis.
My main concern is that the findings of improved symptoms in some cases in this cohort study are rather weak and require further investigation with stronger designs to eliminate confounding factors. I would hesitate to describe this as a 'groundbreakting report." See also my comments in the attached pdf file.

Author Response
Comments[The researchers used media conditioned stem cells from human exfoliated deciduous teeth to treat symptoms of amyotrophic lateral sclerosis.
My main concern is that the findings of improved symptoms in some cases in this cohort study are rather weak and require further investigation with stronger designs to eliminate confounding factors. I would hesitate to describe this as a 'groundbreakting report." See also my comments in the attached pdf file.]
Response[
Thank you for your insightful comments and suggestions on our manuscript titled "A Retrospective Cohort Analysis of the Safety and Efficacy of Conditioned Media from Human Exfoliated Deciduous Tooth Stem Cells in the Treatment of Amyotrophic Lateral Sclerosis (ALS)." We appreciate your feedback and the opportunity to refine our manuscript.Please refer to the following comments for detailed responses, and see the file's revision history for the changes made. Some expressions have been modified to improve the quality of the paper.
We understand the concern regarding the strength of the findings on symptom improvement in some cases within this cohort study. In response, we have revised the phrase "groundbreaking report" to more accurately reflect the preliminary nature of our findings and the need for further investigation.
Original Text:
"... this can be considered a groundbreaking report."
Revised Text:
"... this represents a promising preliminary finding."
We agree that further investigation with more robust study designs is necessary to eliminate confounding factors and validate these initial observations. We have incorporated this revised phrasing into the manuscript and ensured that the discussion emphasizes the exploratory nature of our study and the need for future research. Therefore, we have added the following sentence to the limitations section.
(Finally, although the efficacy results of this study are preliminary, they suggest potential benefits that require further investigation. We emphasize that the efficacy findings are exploratory and suggest future studies with larger cohorts to validate these results.)
Additionally, we have reviewed and addressed all the comments provided in the attached PDF file to further improve the manuscript.Some expressions have also been reviewed.
Thank you again for your valuable feedback.
Sincerely,
]

Reviewer 2 Report
Comments and Suggestions for Authors
The manuscript by Seta et al. represents a retrospective cohort analysis of the safety and efficacy of conditioned media from human exfoliated deciduous tooth stem cells in the treatment of amyotrophic lateral sclerosis (ALS). Unfortunately, the number of subjects involved was limited to 25, and therefore only some data on safety were obtained, while the results with efficacy remained unclear. In addition, the authors present their interpretation on LC-MS data for SHED-CMs, which is also unclear, while primary data are not provided. Thus, it is possible to conclude that this manuscript only contains safety data obtained in a small cohort. In my opinion, this is not sufficient for publication in Biomedicines.
Author Response
Comments:[The manuscript by Seta et al. represents a retrospective cohort analysis of the safety and efficacy of conditioned media from human exfoliated deciduous tooth stem cells in the treatment of amyotrophic lateral sclerosis (ALS). Unfortunately, the number of subjects involved was limited to 25, and therefore only some data on safety were obtained, while the results with efficacy remained unclear. In addition, the authors present their interpretation on LC-MS data for SHED-CMs, which is also unclear, while primary data are not provided. Thus, it is possible to conclude that this manuscript only contains safety data obtained in a small cohort. In my opinion, this is not sufficient for publication in Biomedicines. ]
Response
Thank you for your valuable feedback on our manuscript titled "A Retrospective Cohort Analysis of the Safety and Efficacy of Conditioned Media from Human Exfoliated Deciduous Tooth Stem Cells in the Treatment of Amyotrophic Lateral Sclerosis (ALS)." We appreciate the opportunity to address the concerns raised and provide clarifications.
Please refer to the following comments for detailed responses, and see the file's revision history for the changes made. Some expressions have been modified to improve the quality of the paper.
1. Number of Subjects and Data on Efficacy: We acknowledge that the limited sample size of 24 subjects may affect the generalizability of our findings. Our primary objective was to evaluate the safety profile of SHED-CM in patients with ALS , and our findings indicated a favorable safety profile. However, we also observed some indications of efficacy, such as improvements in ALSFRS-R scores and muscle strength in certain patients. While these efficacy results are preliminary, they suggest potential benefits that warrant further investigation. We will emphasize the exploratory nature of our efficacy findings and propose future studies with larger cohorts to validate these results.
2. LC-MS Data Interpretation: We understand the concern regarding the clarity of the LC-MS data interpretation for SHED-CM. We have reviewed the relevant sections and will revise them to provide a more detailed explanation of our findings. Additionally, we will include the primary LC-MS data publicly available in the supplementary materials to enhance transparency and allow for a more comprehensive evaluation by readers.
We have added the corrected LC-MS data from column 266.
3. Sufficiency for Publication: While we recognize the limitations of our study, we believe that the safety data we have obtained are valuable and contribute to the understanding of SHED-CM as a potential therapeutic option for ALS. We propose to enhance the manuscript by incorporating the suggested revisions, providing additional context on the exploratory efficacy findings, and including primary LC-MS data. We hope these improvements will address the concerns raised and demonstrate the significance of our work for consideration in Biomedicines.
The following sentence has been added to the limitations section:
Finally, although the efficacy results of this study are preliminary, they suggest potential benefits that require further investigation. We emphasize that the efficacy findings are exploratory and suggest future studies with larger cohorts to validate these results.
Thank you for your time and consideration. We look forward to your feedback on our revised manuscript.
Sincerely,
Seta et al.

Reviewer 3 Report
Comments and Suggestions for Authors
Authors describe the safety and efficacy of SHED-CM in ALS treatment. I agree with authors comments that three drugs used for the treatment of ALS approved from the FDA can only slow disease progression to tracheostomy for 3–6 months. I think that this paper may have some value to be published in Biomedicines, but there are some major comments.
Comments to the authors.
Major comments: 
Authors mention that “This study was approved by the Ethics Committee of the Japan Registry of Clinical Trials (jRCT1031230719)”. Why do authors design this study as retrospective study? Recently, clinical trial must be a prospective study at least as “Open label prospective study”.
L85; “at our hospital.” Are authors belong to Hospital? not a clinic?
L87; This was a retrospective cohort study including patients diagnosed with ALS who visited our hospital between January 1, 2022, and November 30, 2023. Are all ALS patients who visited their hospital included in this study? The inclusion and exclusion criteria are not clear.
What are the five authors roles for example clinical evaluation, data analysis, SHED-CM infusion? I think that the members of clinical evaluation and data analysis must be independent. Are there no Clinical Research Coordinators (CRC)?
Minor comments;
Are authors planning the randomized clinical trial using SHED-CM for ALS?
Author Response
Thank you for your thoughtful comments and suggestions regarding our manuscript titled "A Retrospective Cohort Analysis of the Safety and Efficacy of Conditioned Media from Human Exfoliated Deciduous Tooth Stem Cells in the Treatment of Amyotrophic Lateral Sclerosis (ALS)." We appreciate the opportunity to clarify and address the points raised.
Major comments: [Authors mention that “This study was approved by the Ethics Committee of the Japan Registry of Clinical Trials (jRCT1031230719)”. Why do authors design this study as retrospective study? Recently, clinical trial must be a prospective study at least as “Open label prospective study”.]
Response[
Study Design: Our primary goal was to conduct an initial safety assessment of SHED-CM in patients with ALS using existing real-world clinical data. Given the exploratory nature of this research and the limited availability of initial data, a retrospective design was chosen to quickly gather preliminary insights. We acknowledge the importance of prospective studies and plan to conduct a more robust open-label prospective study to further validate our findings. We will revise the manuscript to clearly state the rationale for our study design and outline our future plans for prospective studies.
The following sentence has been added to the limitations section:
Finally, although the efficacy results of this study are preliminary, they suggest potential benefits that require further investigation. We emphasize that the efficacy findings are exploratory and suggest future studies with larger cohorts to validate these results.]
comments: [L85; “at our hospital.” Are authors belong to Hospital? not a clinic?]
Response[We are belong to a clinic and this study was conducted at a clinic. The part pointed out was incorrect, so I have corrected it to "clinic."]
comments: [L87; This was a retrospective cohort study including patients diagnosed with ALS who visited our hospital between January 1, 2022, and November 30, 2023. Are all ALS patients who visited their hospital included in this study? The inclusion and exclusion criteria are not clear.]
Response[As you pointed out, the inclusion and exclusion criteria were missing.We have added the inclusion and exclusion criteria to the Materials and Methods section.
We have revised the manuscript to include the following criteria:
Inclusion criteria
Patients who meet all of the following conditions are eligible for the study:
- Patients who visited our hospital between January 1, 2022, and November 30, 2023, and have been diagnosed with ALS.
- Males and females aged 16 to 90 years.
- ALSFRS-R score of 12 points or more at the first visit.
Exclusion criteria
Patients who meet any of the following criteria will not be included in this study.
- Patients with a tracheotomy.
- Patients who have expressed their refusal to participate in this study.]
comments: [What are the five authors roles for example clinical evaluation, data analysis, SHED-CM infusion? I think that the members of clinical evaluation and data analysis must be independent. Are there no Clinical Research Coordinators (CRC)?]
Response[Authors' Roles and Independence:
We acknowledge the concern about the need for independence between clinical evaluation and data analysis. Due to the limited resources and the exploratory nature of this initial study, same Dr was responsible for both clinical evaluation and data analysis. In future studies, we plan to ensure greater independence between these roles to enhance objectivity.
Clinical Research Coordinators (CRCs):
Our study did not involve Clinical Research Coordinators (CRCs) due to resource constraints. We recognize the value of CRCs in ensuring rigorous data collection and patient follow-up. Future studies will incorporate CRCs to strengthen the study design and data integrity.]
Minor comments;
Response[Are authors planning the randomized clinical trial using SHED-CM for ALS?
We are, of course, planning prospective studies and randomized clinical trials using SHED-CM for ALS, and we are very eager to see these come to fruition]
